# Dynamic Voting for Efficient Reasoning in Large Language Models

**Mingfeng Xue**♠   **Dayiheng Liu**[*]  **Wenqiang Lei**♠   **Xingzhang Ren**
**Baosong Yang**   **Jun Xie**   **Yidan Zhang**♠   **Dezhong Peng**♠   **Jiancheng Lv**♠
♠ College of Computer Science, Sichuan University
♣ Engineering Research Center of Machine Learning and Industry Intelligence
mingfengxue@stu.scu.edu.cn    losinuris@gmail.com

## Abstract

Multi-path voting methods like self-consistency have been used to mitigate reasoning errors in large language models caused by factual errors and illusion generation. However, these methods require excessive computing resources as they generate numerous reasoning paths for each problem. Our experiments show that on the arithmetic reasoning task, SVAMP, half of the problems fail to obtain noticeable accuracy gains when voting with more than three paths. In this paper, we propose a novel multi-path voting technique called Dynamic Voting, which effectively reduces the number of reasoning paths during multi-path voting while preserving accuracies by applying early exiting for problems that large language models can confidently solve. Experimental evaluations on arithmetic, commonsense, and symbolic reasoning tasks under few-shot and zero-shot settings demonstrate that Dynamic Voting achieves comparable accuracies employing significantly fewer reasoning paths. Notably, one of our Dynamic Voting strategies outperforms self-consistency using only 24.7% of the number of paths on the LetterConcat task in the few-shot setting. Furthermore, Dynamic Voting showcases strong robustness of thresholds. It also demonstrates excellent generalizability when combined with other voting techniques, different models, and diverse prompts.

## 1 Introduction

Prominent large language models (LLMs) like GPT-3 (Brown et al., 2020; Ouyang et al., 2022), Chinchilla (Hoffmann et al., 2022), and PaLM (Chowdhery et al., 2022; Chung et al., 2022) demonstrate exceptional performance in natural language processing tasks. Despite the success, their performance on reasoning-heavy tasks, such as mathematics, common sense, and logical reasoning tasks, remains limited when solely increasing

_______________
[*] Corresponding author.

the model size (Rae et al., 2021). To address this issue, Wei et al. (2022) propose chain of thoughts (CoT) to guide LLMs to reason step by step, which however suffers from factual errors and hallucinations in reasoning paths (Cobbe et al., 2021; Wang et al., 2022b; Li et al., 2022; Weng et al., 2022; Ye and Durrett, 2022). To mitigate the errors of CoT, researchers propose multi-path voting methods, including self-consistency (Wang et al., 2022b), DIVERSE (Li et al., 2022), and rationale-augmented ensembles (Wang et al., 2022a), which generate multiple reasoning paths and aggregate the paths with voting. However, multiple reasoning paths lead to significant increases in computational resources. For instance, self-consistency generates 40 reasoning paths per question, while DIVERSE employs 100, resulting in tens of times more computational effort.

This paper aims to address the following problem: *how to achieve comparable accuracy in multi-path voting using significantly fewer computational resources?* One intuition is that simple problems can be solved with fewer reasoning paths and tilting computational resources to complex problems will effectively improve the overall performance of reasoning. In our empirical investigation on an arithmetic reasoning task, SVAMP (Patel et al., 2021), 57.5% of the problems demonstrate a remarkable level of accuracy, reaching 95.7%, with only three unanimous reasoning paths. And augmenting the reasoning paths on these problems does not significantly improve the accuracy. This shows that generating more paths on half of the problems in SVAMP is a waste of computational resources with negligible benefit. A more detailed analysis of resource wastage can be found in Appendix A.

Inspired by Early Exiting (Viola and Jones, 2001), which prioritizes computational resources to critical features to minimize computational consumption, we propose *Dynamic Voting* to reduce the number of reasoning paths while maintaining

comparable accuracies in reasoning with LLMs. The core idea behind Dynamic Voting is to apply early exiting for problems that the LLMs can confidently solve with a handful of paths, thus avoiding unnecessary generation of numerous paths.

A key challenge in Dynamic Voting lies in the identification of problems that the LLMs can confidently solve. Self-consistency establishes a robust association between voting consistency and accuracy, which aligns with our finding that a few unanimous paths on SVAMP lead to high accuracy. Thus, we adopt voting consistency as the confidence of the LLMs in solving problems and validate it through a range of experiments.

Dynamic Voting is a simple and effective method that involves multiple rounds of voting with corresponding consistency thresholds. Initially, the LLM generates a few paths and votes on them. If the voting consistency threshold is reached, the voting concludes and the current voting result is outputted. Otherwise, the LLM generates another reasoning path and all the generated paths are used for the subsequent round of voting. This iterative process continues until the threshold is reached or the number of paths reaches a preset maximum. Figure 1 illustrates an example of the process.

In practice, we introduce two Dynamic Voting strategies: Confidence-based Dynamic Voting (CDV) and Percentage-based Dynamic Voting (PDV). CDV employs a fixed threshold in each round, offering a straightforward approach. However, it is limited by the need for prior knowledge regarding the probability of correctly solving the problem. For example, when the probability of correct reasoning is 0.5, setting a threshold of 0.9 renders early exiting unlikely. To overcome this limitation, PDV applies early exiting to a fixed proportion of questions that obtain the highest consistency in the current voting round, offering greater flexibility and adaptability in determining when to terminate the voting process. We evaluate these strategies using the Openai GPT-3.5 model[1] in few-shot and zero-shot settings. Following Wei et al. (2022) and Wang et al. (2022b), we conduct experiments on Arithmetic Reasoning (GSM8K (Cobbe et al., 2021), SVAMP (Patel et al., 2021)), Commonsense Reasoning (CSQA (Talmor et al., 2019), StrategyQA (Geva et al., 2021)), and Symbolic Reasoning (LetterConcat (Wei et al., 2022)) tasks.

Dynamic Voting achieves comparable accuracies using significantly fewer reasoning paths than self-consistency[2]. Across the five evaluated datasets, Dynamic Voting achieves a comparable average accuracy to self-consistency while employing less than 45% of the reasoning paths. Specifically, in the few-shot setting on LetterConcat, CDV outperforms self-consistency using 24.7% of the reasoning paths. Moreover, Dynamic Voting demonstrates substantial accuracy improvements under reduced path constraints. When both are limited to 25% reasoning path usage[3], Dynamic Voting achieves an average accuracy gain of 1.6 and 4.7 over self-consistency in the few-shot and zero-shot settings, respectively. Remarkably, PDV improves by 12.7 on LetterConcat in the zero-shot setting. Furthermore, Dynamic Voting attains higher accuracies when employing comparable paths. On GSM8K, CDV and PDV achieve improvements of 1.9 and 1.0, respectively, compared to the highest accuracy of self-consistency in the zero-shot setting with less than 100% reasoning path usage.

In addition to its resource efficiency, Dynamic Voting exhibits several excellent features. Dynamic Voting demonstrates stronger robustness of thresholds than self-consistency, providing flexibility in setting appropriate threshold levels. Moreover, Dynamic Voting performs well when combined with other voting techniques, different LLMs, and diverse prompts.

## 2 Related Works

### 2.1 CoT Reasoning with LLMs

Large language models have achieved remarkable success in natural language processing tasks (Srivastava et al., 2022), even surpassing human-level performance in some cases (Brown et al., 2020; Zhang et al., 2022a; Scao et al., 2022; Ouyang et al., 2022; Hoffmann et al., 2022; Chowdhery et al., 2022; Chung et al., 2022). However, the ability of these models to solve complex reasoning problems does not improve significantly by simply increasing model size (Rae et al., 2021). To address this, Wei et al. (2022) propose chain of thoughts (CoT), which is a sequence of intermediate steps (Ling et al., 2017), to assist the model in reasoning step by step.

---

[1]We utilize the GPT-3.5-turbo-0301 API (https://platform.openai.com/docs/models/gpt-3-5) since it is fee friendly and not updated iteratively like GPT-3.5-turbo.

[2]We consider self-consistency as the basic voting technique due to its computational simplicity.

[3]We adopt the same path setting as self-consistency, taking forty paths per question as the maximum (100%).

Several approaches have been proposed to improve the performance of CoT. Some researchers attempt to break down complex problems into sub-problems, which are then solved sequentially (Zhou et al., 2022; Drozdov et al., 2022). Others explore the use of additional tools such as code, retrieval, and symbolic languages to provide extra information (Yao et al., 2022; Cheng et al., 2022; Chen et al., 2022; Gao et al., 2022; Schick et al., 2023). A simpler yet effective approach is self-consistency (Wang et al., 2022b), which generates multiple reasoning paths and yields the final output through majority vote. Another approach involves calibrating the reasoning paths using an additional verifier (Li et al., 2022; Ye and Durrett, 2022). However, calibrating approaches require training additional verifiers and exhibit performance degradation on out-of-distribution problems.

Multi-path voting is effective, but comes with the drawback of causing a substantial increase in computational resources due to multi-path generation. To address this issue, this paper aims to achieve a comparable accuracy with significantly fewer reasoning paths in reasoning with LLMs.

## 2.2 Early Exiting

Early Exiting optimizes inference and resource utilization by interrupting computation when adequate results are produced. To accelerate real-time face detection, Viola and Jones (2001) propose to select a few crucial features from a vast number of possible features for further processing. Similarly, Persin (1994) and Cambazoglu et al. (2010) identify potentially highly ranked documents beforehand to improve retrieval efficiency. In addition, some researchers attempt to reduce the number of model layers involved in computation by relying on intermediate layer outputs through entropy-based (Teerapittayanon et al., 2016; Xin et al., 2020; Liu et al., 2020), confidence-based (Kaya et al., 2019; Schwartz et al., 2020), and patience-based (Zhou et al., 2020) indicators. Drawing on these insights, we propose Dynamic Voting that aims to conserve computational resources in multi-path voting by performing early exiting to prevent unnecessary path generation. To the best of our knowledge, this is the first work to apply early exiting to voting with LLMs.

## 3 Dynamic Voting

This section provides a comprehensive exposition of our proposed method, Dynamic Voting. We present an overview of the method in Section 3.1. We then introduce two Dynamic Voting strategies, Confidence-based Dynamic Voting and Percentage-based Dynamic Voting in Section 3.2 and 3.3.

### 3.1 Overall Process

As stated in Section 1, the consistency of the votes has a strong correlation with the reasoning accuracy. Thus, we estimate the confidence of the LLMs using consistency. How to reasonably quantify voting consistency is one of the crucial factors in the success of Dynamic Voting. In this paper, we simply use the proportion of the majority votes in each voting round as consistency and experimentally demonstrate the effectiveness of this plain scheme.

Figure 1 illustrates an example of the overall process of Dynamic Voting, which involves multiple rounds of generation and voting progress. We first set the consistency threshold $t$, the initial number of votes $v$, and the maximum number of votes $V$. For a problem $p$, we use the LLM to generate $v$ reasoning paths and vote on these paths. If the voting consistency reaches $t$, the voting result is considered the final reasoning output, and the generation and voting process is exited. Otherwise, Dynamic Voting will proceed to the next round where the LLM generates another path and the voting process is carried out on all the $v + 1$ generated paths. This generation and voting process continues until the voting consistency reaches $t$ or until $V$ paths are generated and voted on.

It is worth noting that varied thresholds result in different accuracies and computational resource consumption. A loose threshold allows more questions to exit early, which saves more computational resources but reduces accuracy. Conversely, strict thresholds lead to more computational resource consumption and higher accuracy.

### 3.2 Confidence-based Dynamic Voting

In Confidence-based Dynamic Voting (CDV), a consistency threshold $t_c$ is set in each round of voting. If the proportion of the majority votes in the current round reaches the set threshold, the result is considered valid, and early exiting is performed. For example, there are 10 paths in the $k$-th round of voting and $t_c = 0.5$, the voting ends if the number of paths to an answer reaches 5. Otherwise, the pro-

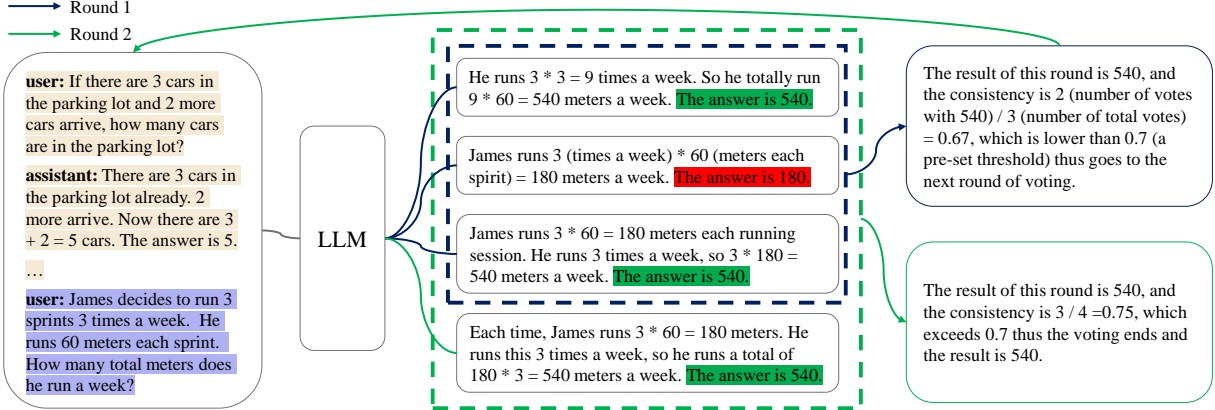

Figure 1: The overall process of Dynamic Voting. The figure shows an example of two rounds of voting with the consistency threshold set to 0.7. In Round 1, the voting consistency does not reach the pre-set threshold, so the voting proceeds to Round 2. In Round 2, the LLM generates an additional path to participate in the voting and the consistency reaches the threshold, upon which the voting ends and the voting result is output. In practice, voting will continue until the consistency threshold is reached or the number of generated paths reaches a predefined maximum.

cess continues to the next round of path generation and voting.

In general, achieving high confidence in later rounds of the Dynamic Voting process is less likely as those questions that the LLM can solve confidently are usually exited in the early stages. An ideal option for designing confidence-based thresholds would be to gradually decrease the threshold value as the number of rounds increases. However, in this paper, we opt for a simple approach of setting a constant threshold value for all rounds to avoid complex fine-grained threshold tuning.

### 3.3 Percentage-based Dynamic Voting

One challenge in effectively utilizing CDV is the selection of an appropriate threshold based on prior knowledge, as discussed in Section 1. To facilitate a simpler and more efficient application of Dynamic Voting, we propose a strategy that automatically adjusts the consistency threshold when dealing with batch problems, called Percentage-based Dynamic Voting (PDV). In PDV, we set a percentage threshold $t_p$, and then the questions with the top $t_p$% of consistencies in each round will be applied early exiting. The percentage threshold is solely employed to control the proportion of questions exiting early in each round, eliminating the need for any prior knowledge about the LLMs or the questions themselves.

PDV can handle batch problems quite efficiently as it ensures that a certain percentage of questions always exit in each voting round. The disadvantage of PDV is that it does not work on individual questions well. It is noteworthy that in PDV,

an increased threshold implies the early exit of a greater number of questions each round, which corresponds to lower accuracy and reduced computational resource consumption.

## 4 Experiments

### 4.1 Datasets

Following Wei et al. (2022) and Wang et al. (2022b), we conduct experiments on Arithmetic, Commensense, and Symbolic reasoning tasks. It is worth noting that GPT-3.5 with self-consistency is capable of achieving close to 100% accuracy on certain tasks such as MultiArith (Roy and Roth, 2015), ARC (Clark et al., 2018), and CoinFlip (Wei et al., 2022). Consequently, we carefully select several representative datasets that are moderately challenging for GPT-3.5.

For the Arithmetic Reasoning task, we utilize GSM8K (Cobbe et al., 2021) and SVAMP (Patel et al., 2021), two challenging sets of math word problems. For Commensense Reasoning, we employ CSQA (Talmor et al., 2019) and StrategyQA (Geva et al., 2021), both of which are considered to be highly challenging question answering benchmarks that require a general understanding of common sense. Finally, for Symbolic Reasoning, we take the LetterConcat task (Wei et al., 2022), in which GPT-3.5 is required to concatenate the last letters of four given words.

In our experiments, we use the test split for GSM8K and SVAMP. For CSQA, we use the dev split since the labels of the test set are not available. As for LetterConcat, we construct the test set by

randomly generating 1000 examples using the 1000 most commonly used surnames and first names from Name Census[4]. The settings for the aforementioned four datasets align with those employed by Wang et al. (2022b). Regarding StrategyQA, we randomly sample 1000 questions from the train-filtered set because the evaluation on the test set necessitates the use of the StrategyQA Leaderboard[5]. Multiple evaluations on various thresholds are required in this study and frequent submissions are not allowed in the Leaderboard.

## 4.2 Prompts and Instructions

We conduct experiments on the GPT-3.5 model[6] in both few-shot and zero-shot settings. To form the prompts for the few-shot setting, we use the same demonstrations as proposed by Wang et al. (2022b), with minor modifications made to create dialog histories that GPT-3.5 can leverage. In the zero-shot setting, we provide one-sentence instructions that set constraints on the desired word count and the format of the final answer. The complete prompts and instructions used on the five evaluated datasets are provided in Appendix E.

## 4.3 Hyper Parameters

We compare the performance of Dynamic Voting and Self-consistency under various path usage constraints, including 25%, 50%, 75%, and 100%. Specifically, following Wang et al. (2022b), the maximum path constraint (100%) is defined as voting with forty votes per question.

For Self-consistency, constraints result in a reduced number of votes per question. For example, a usage constraint of 25% implies the use of 10 paths per question, while a constraint of 50% corresponds to 20 paths, and so forth. In the case of Dynamic Voting, the initial round commences with 3 votes. The constraints in Dynamic Voting indicate the percentage of used paths on a dataset relative to the maximum path constraint.

To determine the appropriate thresholds for Dynamic Voting under different reasoning path usage constraints, we follow the methodology presented by Kaya et al. (2019) and use a validation set comprising 100 randomly selected samples from the training set. In CDV, the threshold search range is

set between 0.1 and 0.95, with a granularity of 0.05. For PDV, the range is set between 1 and 30, with a granularity of 1. Notably, the threshold searching necessitates only one generation of 40 paths for 100 examples on each dataset, thus the searching computational overhead is trivial. The searched thresholds can be found in Appendix B.

In accordance with Wang et al. (2022b), we incorporate temperature sampling to generate diverse reasoning paths. During decoding, we set the temperature to 1.5 without top-k/top-p truncation.

## 4.4 Experimental Results

### 4.4.1 Fewer Paths for Comparable Accuracy

Table 1 and Figure 2 present the experimental results on various reasoning path usage constraints and the following conclusions can be drawn:

**1) Dynamic Voting achieves comparable accuracies to self-consistency with significantly fewer reasoning paths.** In Table 1, in the few-shot and zero-shot settings, Dynamic Voting requires only 25% and 50% of the reasoning path usages, respectively, while maintaining average accuracies on par with self-consistency. In particular, the CDV in the zero-shot setting on StrategyQA and the PDV in the few-shot setting on LetterConcat meet the accuracy of self-consistency using less than 25% of the paths. In Figure 2, Dynamic Voting achieves the highest accuracy of SC using less than 60% of the paths. This highlights the potential of Dynamic Voting to conserve substantial computational resources compared to self-consistency.

**2) Dynamic Voting significantly outperforms self-consistency on all tasks under a limited reasoning path usage.** In Table 1, When the reasoning path usage constraint is set to 25%, Dynamic Voting surpasses self-consistency on all five tasks. On average, CDV and PDV exhibit performance improvements of 1.6 and 1.6 in the few-shot setting, and 4.7 and 4.8 in the zero-shot setting, respectively, compared to self-consistency. In Figure 2, Dynamic Voting consistently achieves higher accuracies than SC when the path usage limit is below 80%. These results indicate that Dynamic Voting offers a substantial advantage over self-consistency in scenarios with constrained computational resources.

**3) Dynamic Voting reduces reasoning path usage with negligible performance degradation under extremely strict thresholds.** When applying strict thresholds in Dynamic Voting, only a

---

[4]https://namecensus.com/.
[5]https://leaderboard.allenai.org/strategyqa/submissions/public.
[6]We utilize the GPT-3.5-turbo-0301 API (https://platform.openai.com/docs/models/gpt-3-5).

| Task | Method | fewshot | | | | zeroshot | | | |
|---|---|---|---|---|---|---|---|---|---|
| | | $\leq 25\%$ | $\leq 50\%$ | $\leq 75\%$ | $\leq 100\%$ | $\leq 25\%$ | $\leq 50\%$ | $\leq 75\%$ | $\leq 100\%$ |
| GSM8K | SC | $82.6_{(25.0)}$ | $86.1_{(50.0)}$ | $87.3_{(75.0)}$ | $87.9_{(100.0)}$ | $80.2_{(25.0)}$ | $84.0_{(50.0)}$ | $85.1_{(75.0)}$ | $85.4_{(100.0)}$ |
| | CDV | $\mathbf{86.5}_{(25.0)}$ | $\underline{\mathbf{87.9}}_{(42.9)}$ | $\underline{\mathbf{87.9}}_{(52.3)}$ | $\underline{87.9}_{(52.3)}$ | $\mathbf{82.7}_{(28.4)}$ | $\mathbf{85.1}_{(46.6)}$ | $\mathbf{85.3}_{(60.4)}$ | $\underline{85.4}_{(68.0)}$ |
| | PDV | $\mathbf{85.8}_{(24.7)}$ | $\mathbf{87.6}_{(43.1)}$ | $\underline{\mathbf{87.9}}_{(65.5)}$ | $\underline{87.9}_{(85.9)}$ | $\mathbf{84.2}_{(25.6)}$ | $\underline{\mathbf{85.4}}_{(43.1)}$ | $\underline{\mathbf{85.7}}_{(58.1)}$ | $\underline{85.4}_{(85.9)}$ |
| SVAMP | SC | $84.5_{(25.0)}$ | $86.3_{(50.0)}$ | $86.3_{(75.0)}$ | $86.7_{(100.0)}$ | $82.9_{(25.0)}$ | $85.9_{(50.0)}$ | $87.6_{(75.0)}$ | $87.4_{(100.0)}$ |
| | CDV | $\mathbf{86.0}_{(23.6)}$ | $86.1_{(31.4)}$ | $86.1_{(37.5)}$ | $86.1_{(37.5)}$ | $\mathbf{85.1}_{(23.8)}$ | $\mathbf{87.1}_{(40.7)}$ | $87.2_{(57.0)}$ | $87.2_{(57.0)}$ |
| | PDV | $\mathbf{86.2}_{(25.6)}$ | $\underline{\mathbf{86.8}}_{(47.2)}$ | $\underline{\mathbf{86.7}}_{(58.1)}$ | $\underline{86.7}_{(85.9)}$ | $\mathbf{86.4}_{(24.7)}$ | $\underline{\mathbf{87.8}}_{(43.2)}$ | $\underline{87.4}_{(65.5)}$ | $\underline{87.4}_{(85.9)}$ |
| CSQA | SC | $80.1_{(25.0)}$ | $80.5_{(50.0)}$ | $80.6_{(75.0)}$ | $81.0_{(100.0)}$ | $72.2_{(25.0)}$ | $75.8_{(50.0)}$ | $75.6_{(75.0)}$ | $75.8_{(100.0)}$ |
| | CDV | $\mathbf{80.5}_{(23.3)}$ | $\mathbf{80.8}_{(37.1)}$ | $\mathbf{80.8}_{(40.2)}$ | $80.8_{(40.2)}$ | $\mathbf{74.0}_{(18.4)}$ | $\underline{75.8}_{(39.6)}$ | $\underline{\mathbf{75.9}}_{(64.4)}$ | $75.7_{(93.4)}$ |
| | PDV | $\mathbf{80.3}_{(21.9)}$ | $\underline{\mathbf{81.0}}_{(47.1)}$ | $\underline{\mathbf{81.0}}_{(65.5)}$ | $\underline{81.0}_{(85.9)}$ | $\mathbf{75.6}_{(22.5)}$ | $\underline{\mathbf{76.0}}_{(43.1)}$ | $\underline{75.8}_{(65.5)}$ | $\underline{75.8}_{(85.9)}$ |
| StrategyQA | SC | $75.7_{(25.0)}$ | $77.0_{(50.0)}$ | $77.2_{(75.0)}$ | $76.7_{(100.0)}$ | $71.8_{(25.0)}$ | $73.5_{(50.0)}$ | $74.2_{(75.0)}$ | $73.7_{(100.0)}$ |
| | CDV | $\mathbf{76.3}_{(22.0)}$ | $\underline{76.9}_{(43.6)}$ | $\underline{76.9}_{(54.8)}$ | $\underline{\mathbf{76.9}}_{(55.1)}$ | $\underline{\mathbf{73.7}}_{(22.5)}$ | $73.6_{(38.6)}$ | $\underline{73.8}_{(64.3)}$ | $\underline{\mathbf{73.8}}_{(67.2)}$ |
| | PDV | $\mathbf{76.4}_{(22.5)}$ | $\underline{\mathbf{77.4}}_{(36.9)}$ | $76.7_{(65.5)}$ | $76.7_{(85.9)}$ | $73.0_{(22.5)}$ | $\underline{\mathbf{74.1}}_{(47.2)}$ | $73.7_{(58.1)}$ | $73.7_{(85.9)}$ |
| LetterConcat | SC | $85.0_{(25.0)}$ | $86.2_{(50.0)}$ | $86.8_{(75.0)}$ | $86.7_{(100.0)}$ | $62.4_{(25.0)}$ | $79.8_{(50.0)}$ | $86.3_{(75.0)}$ | $89.9_{(100.0)}$ |
| | CDV | $\mathbf{86.4}_{(23.4)}$ | $\mathbf{86.4}_{(31.5)}$ | $86.4_{(31.5)}$ | $86.4_{(31.5)}$ | $\mathbf{78.7}_{(30.9)}$ | $\mathbf{84.8}_{(48.5)}$ | $89.7_{(72.4)}$ | $\underline{89.9}_{(97.0)}$ |
| | PDV | $\mathbf{87.0}_{(24.7)}$ | $\mathbf{86.8}_{(43.2)}$ | $86.7_{(65.5)}$ | $86.7_{(85.9)}$ | $\mathbf{75.1}_{(24.7)}$ | $\mathbf{89.3}_{(47.2)}$ | $\mathbf{89.8}_{(65.5)}$ | $\underline{89.9}_{(85.9)}$ |
| AVG | SC | $81.6_{(25.0)}$ | $83.3_{(50.0)}$ | $83.7_{(75.0)}$ | $83.9_{(100.0)}$ | $74.2_{(25.0)}$ | $79.9_{(50.0)}$ | $81.7_{(75.0)}$ | $82.4_{(100.0)}$ |
| | CDV | $\mathbf{83.2}_{(23.5)}$ | $\mathbf{83.8}_{(37.6)}$ | $\mathbf{83.8}_{(43.7)}$ | $83.8_{(43.7)}$ | $\mathbf{78.9}_{(24.8)}$ | $\mathbf{81.3}_{(42.9)}$ | $\mathbf{82.3}_{(63.5)}$ | $82.3_{(76.7)}$ |
| | PDV | $\mathbf{83.2}_{(23.9)}$ | $\underline{\mathbf{84.0}}_{(43.6)}$ | $\underline{\mathbf{83.9}}_{(64.2)}$ | $\underline{83.9}_{(85.9)}$ | $\mathbf{79.0}_{(24.0)}$ | $\underline{\mathbf{82.4}}_{(44.6)}$ | $\underline{\mathbf{82.4}}_{(62.4)}$ | $82.3_{(85.9)}$ |

Table 1: The accuracies of Self-consistency and Dynamic Voting under different reasoning path usage constraints. "SC" means self-consistency. $\leq 25\%$ indicates that when applying the threshold, less than 25% of the paths are used on the validation set compared with the maximum path constraint (forty votes per question). Since the threshold is searched from the validation set, the actual usage on the test set may be slightly higher than the constraint. The values in brackets indicate the actual usage on the test set (%). The accuracies exceeding SC under each constraint are marked in **bold**. The accuracies reaching SC (40-votes) are marked underlined.

small number of questions exit early in each voting round. The $\leq 100\%$ results in Table 1 exemplify this situation[7]. Despite these stringent thresholds, Dynamic Voting achieves a reduction in reasoning path usage with an average accuracy loss of no more than 0.1% in both few-shot and zero-shot settings. In particular, the CDV in the few-shot setting uses only 43.7% of the usage while maintaining high accuracy. These findings emphasize the practical applicability of Dynamic Voting, even if the threshold search phase is omitted and a strict threshold is employed directly.

### 4.4.2 Higher Accuracy on Comparable Paths

In Dynamic Voting, the preset maximum number of votes $V$ also affects the accuracy. We set varying maximums and analyze the effect of different maximums on Dynamic Voting. The thresholds are set to those corresponding to $\leq 50\%$ in Table 1, with which Dynamic Voting achieves accuracies comparable to self-consistency on most tasks.

As shown in Table 2, Dynamic Voting uses more reasoning paths and the accuracies improve as the maximum increases. It is noteworthy that PDV in the zero-shot setting only utilizes additional 3.6%

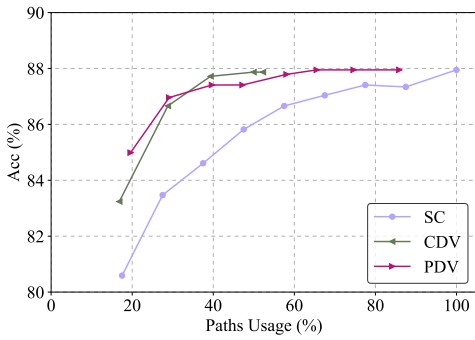

Figure 2: The few-shot results of self-consistency and Dynamic Voting on the GSM8K dataset under different path usage constraints. The zero-shot results and the results on the other datasets are in Appendix C.

of paths to achieve a decent improvement of 0.6 on average accuracy when the maximum is increased from 40 to 100. When $V$ is set to 100, the accuracies of Dynamic Voting on several datasets exceed self-consistency with still fewer reasoning paths, such as GSM8K, SVAMP, and CSQA in the zero-shot setting. This demonstrates that the accuracy can be improved by increasing the maximum number of votes in Dynamic Voting and Dynamic Voting typically outperforms self-consistency with comparable reasoning path usage.

[7]In the experiments, we set $t_c = 0.95$ and $t_p = 1$.

| Task | Method | fewshot | | | | zeroshot | | | |
|---|---|---|---|---|---|---|---|---|---|
| | | max=40 | max=60 | max=80 | max=100 | max=40 | max=60 | max=80 | max=100 |
| GSM8K | SC | 87.9$_{(100.0)}$ | - | - | - | 85.4$_{(100.0)}$ | - | - | - |
| | CDV | 87.9$_{(42.9)}$ | **88.6**$_{(60.2)}$ | **88.7**$_{(77.4)}$ | **88.6**$_{(94.5)}$ | 85.1$_{(46.6)}$ | **86.5**$_{(64.6)}$ | **86.9**$_{(82.4)}$ | **87.2**$_{(99.9)}$ |
| | PDV | 87.6$_{(43.1)}$ | **88.2**$_{(52.1)}$ | **88.3**$_{(66.4)}$ | **88.2**$_{(80.5)}$ | 85.4$_{(43.1)}$ | **86.2**$_{(45.1)}$ | **86.1**$_{(45.6)}$ | **86.1**$_{(45.8)}$ |
| SVAMP | SC | 86.7$_{(100.0)}$ | - | - | - | 87.4$_{(100.0)}$ | - | - | - |
| | CDV | 86.1$_{(31.4)}$ | 86.5$_{(43.0)}$ | 86.6$_{(54.6)}$ | **86.8**$_{(66.1)}$ | 87.1$_{(40.7)}$ | **87.6**$_{(55.8)}$ | **87.5**$_{(70.7)}$ | **88.0**$_{(85.5)}$ |
| | PDV | **86.8**$_{(47.2)}$ | 86.6$_{(37.7)}$ | 86.5$_{(47.3)}$ | 86.6$_{(56.8)}$ | **87.8**$_{(43.2)}$ | **88.3**$_{(45.2)}$ | **88.2**$_{(45.7)}$ | **88.2**$_{(46.1)}$ |
| CSQA | SC | 81.0$_{(100.0)}$ | - | - | - | 75.8$_{(100.0)}$ | - | - | - |
| | CDV | 80.8$_{(37.1)}$ | 80.5$_{(51.4)}$ | 80.6$_{(65.5)}$ | 80.8$_{(79.5)}$ | 75.8$_{(39.6)}$ | **76.4**$_{(52.2)}$ | **77.1**$_{(64.3)}$ | **77.3**$_{(76.3)}$ |
| | PDV | 81.0$_{(47.1)}$ | 80.3$_{(37.4)}$ | 80.4$_{(46.5)}$ | 80.7$_{(55.6)}$ | **76.0**$_{(43.1)}$ | **76.5**$_{(45.1)}$ | **76.7**$_{(45.6)}$ | **76.8**$_{(45.9)}$ |
| StrategyQA | SC | 76.7$_{(100.0)}$ | - | - | - | 73.7$_{(100.0)}$ | - | - | - |
| | CDV | **76.9**$_{(43.6)}$ | 76.7$_{(60.6)}$ | **76.8**$_{(77.3)}$ | **77.3**$_{(94.0)}$ | 73.6$_{(38.6)}$ | 73.5$_{(51.4)}$ | 73.6$_{(63.7)}$ | 73.4$_{(75.6)}$ |
| | PDV | **77.4**$_{(36.9)}$ | 76.6$_{(46.2)}$ | 76.7$_{(57.5)}$ | **77.2**$_{(68.7)}$ | **74.1**$_{(47.2)}$ | **73.9**$_{(50.4)}$ | **73.9**$_{(51.4)}$ | **74.1**$_{(51.8)}$ |
| LetterConcat | SC | 86.7$_{(100.0)}$ | - | - | - | 89.9$_{(100.0)}$ | - | - | - |
| | CDV | 86.4$_{(31.5)}$ | 86.6$_{(44.0)}$ | 86.5$_{(56.5)}$ | 86.5$_{(68.9)}$ | 84.8$_{(48.5)}$ | 86.0$_{(65.8)}$ | 87.2$_{(82.6)}$ | 87.4$_{(99.2)}$ |
| | PDV | **86.8**$_{(43.2)}$ | 86.3$_{(24.1)}$ | 86.3$_{(28.7)}$ | 86.3$_{(33.2)}$ | 89.3$_{(47.2)}$ | 89.7$_{(50.4)}$ | **90.2**$_{(51.4)}$ | **90.1**$_{(51.8)}$ |
| AVG | SC | 83.9$_{(100.0)}$ | - | - | - | 82.4$_{(100.0)}$ | - | - | - |
| | CDV | 83.8$_{(37.6)}$ | 83.9$_{(52.3)}$ | **84.0**$_{(66.9)}$ | **84.1**$_{(81.4)}$ | 81.3$_{(42.9)}$ | 82.0$_{(58.1)}$ | **82.5**$_{(73.0)}$ | **82.7**$_{(87.6)}$ |
| | PDV | **84.0**$_{(43.6)}$ | 83.7$_{(40.1)}$ | 83.8$_{(50.2)}$ | 83.9$_{(60.1)}$ | 82.4$_{(44.6)}$ | **82.9**$_{(47.0)}$ | **83.0**$_{(47.7)}$ | **83.0**$_{(48.0)}$ |

Table 2: The accuracies under different maximum number of votes. Results with accuracy exceeding SC (40-votes) on each dataset are marked in **bold**.

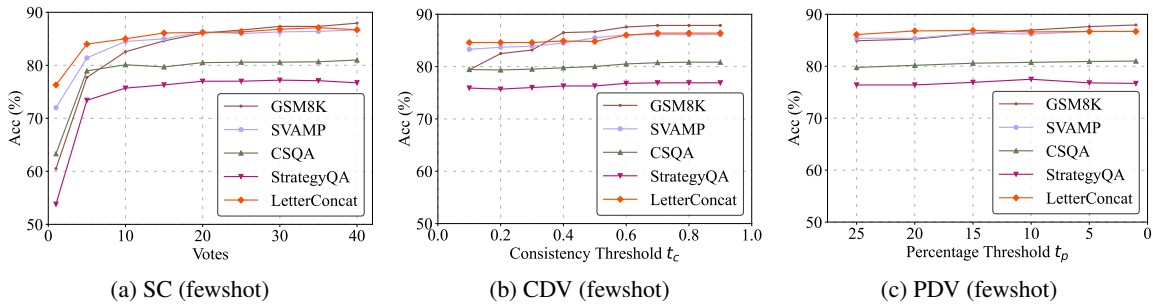

(a) SC (fewshot)  (b) CDV (fewshot)  (c) PDV (fewshot)

Figure 3: The few-shot results of SC and Dynamic Voting under different thresholds. Note that a smaller threshold in PDV implies a stricter exiting condition, that is, a smaller number of samples exiting each round.

## 4.5 Analyses

To get a better insight into Dynamic Voting, we further analyze the applicability of Dynamic Voting in different application scenarios. The high robustness of thresholds provides high fault tolerance in threshold settings, making the method more user-friendly in practical applications. High generalizability when combined with different voting techniques, LLMs, and prompts demonstrates the reliability and expansibility. The analyses are designed for answering the following questions: **Q1:** Does Dynamic Voting show strong robustness of thresholds? **Q2:** Does Dynamic Voting still work with multi-path voting techniques that use Verifiers? **Q3:** Does Dynamic Voting still work on other large language models? **Q4:** Does Dynamic Voting still work across different prompts?

### 4.5.1 Robustness of Thresholds

As detailed in Section 3.1, the selection of thresholds involves a trade-off between accuracy and resource consumption. However, determining the optimal threshold is a challenging task due to the variable probabilities of correct reasoning by LLMs across different problems. The robustness of threshold settings offers greater flexibility, rendering the approaches more user-friendly and practical.

To assess the robustness of threshold settings, we conduct an analysis of the accuracy variations in both self-consistency and Dynamic Voting under different thresholds, and the results are presented in Figure 3. We observe that both self-consistency and Dynamic Voting exhibit a similar trade-off trend: a stricter threshold tends to produce a higher accuracy rate. Notably, the accuracy of Dynamic Voting varies more smoothly with changing thresholds on

| Setting | Method | ≤ 25% | ≤ 50% | ≤ 75% | ≤ 100% |
|---|---|---|---|---|---|
| Few-shot | DIVERSE | 89.2$_{(25.0)}$ | 91.9$_{(50.0)}$ | 92.9$_{(75.0)}$ | 93.0$_{(100.0)}$ |
| | CDV | **92.3**$_{(24.2)}$ | **93.0**$_{(48.6)}$ | **93.0**$_{(49.9)}$ | 93.0$_{(49.9)}$ |
| | PDV | **91.4**$_{(23.1)}$ | **93.0**$_{(47.2)}$ | 93.0$_{(65.5)}$ | 93.0$_{(85.9)}$ |
| Zero-shot | DIVERSE | 85.4$_{(25.0)}$ | 88.7$_{(50.0)}$ | 90.7$_{(75.0)}$ | 91.5$_{(100.0)}$ |
| | CDV | **89.3**$_{(27.3)}$ | **91.5**$_{(46.0)}$ | **91.5**$_{(64.9)}$ | 91.5$_{(64.9)}$ |
| | PDV | **88.5**$_{(23.9)}$ | **91.5**$_{(47.2)}$ | 91.5$_{(65.5)}$ | 91.5$_{(85.9)}$ |

Table 3: Results with DIVERSE on the GSM8K dataset under different resource constraints. The accuracies exceeding DIVERSE under each constraints are marked in **bold** and the accuracies reaching DIVERSE (40-votes) are marked underlined.

| Model | Method | ≤ 25% | ≤ 50% | ≤ 75% | ≤ 100% |
|---|---|---|---|---|---|
| Code | SC | 75.9$_{(25.0)}$ | 78.2$_{(50.0)}$ | 79.5$_{(75.0)}$ | 80.7$_{(100.0)}$ |
| | CDV | **78.4**$_{(25.1)}$ | **80.4**$_{(44.8)}$ | 80.3$_{(58.9)}$ | 80.3$_{(58.9)}$ |
| | PDV | **78.2**$_{(23.9)}$ | **80.6**$_{(43.1)}$ | **80.7**$_{(65.5)}$ | 80.7$_{(85.9)}$ |
| GPT-4 | SC | - | 92.7$_{(50.0)}$ | 94.1$_{(75.0)}$ | 95.2$_{(100.0)}$ |
| | CDV | - | **95.2**$_{(53.9)}$ | **95.2**$_{(53.9)}$ | 95.2$_{(53.9)}$ |
| | PDV | - | **94.6**$_{(53.7)}$ | **95.2**$_{(65.4)}$ | 95.2$_{(81.6)}$ |
| LLaMA-7B | SC | 8.5$_{(25.0)}$ | 11.2$_{(50.0)}$ | 11.4$_{(75.0)}$ | 12.1$_{(100.0)}$ |
| | CDV | **9.0**$_{(26.4)}$ | 10.8$_{(45.2)}$ | **12.7**$_{(68.0)}$ | **12.2**$_{(92.3)}$ |
| | PDV | **9.6**$_{(23.9)}$ | **11.6**$_{(47.2)}$ | **12.6**$_{(74.6)}$ | **12.5**$_{(85.9)}$ |

Table 4: Results on GSM8K using Code-davinci-002 API, GPT-4 API, and LLaMA-7B.

| Prompt | Method | ≤ 25% | ≤ 50% | ≤ 75% | ≤ 100% |
|---|---|---|---|---|---|
| Prompt 2 | SC | 84.4$_{(25.0)}$ | 86.9$_{(50.0)}$ | 87.5$_{(75.0)}$ | 87.3$_{(100.0)}$ |
| | CDV | **86.0**$_{(22.8)}$ | **87.3**$_{(40.4)}$ | 87.3$_{(48.8)}$ | 87.3$_{(48.8)}$ |
| | PDV | **86.9**$_{(23.9)}$ | **87.5**$_{(47.2)}$ | 87.3$_{(74.6)}$ | 87.3$_{(85.9)}$ |
| Prompt 3 | SC | 83.9$_{(25.0)}$ | 86.7$_{(50.0)}$ | 87.6$_{(75.0)}$ | 87.9$_{(100.0)}$ |
| | CDV | **85.8**$_{(23.8)}$ | **87.6**$_{(47.0)}$ | 87.6$_{(55.8)}$ | 87.6$_{(55.8)}$ |
| | PDV | **86.5**$_{(23.9)}$ | **87.6**$_{(47.2)}$ | **87.9**$_{(74.6)}$ | 87.9$_{(85.9)}$ |

Table 5: Results on GSM8K with diverse prompts.

all the five evaluated tasks, suggesting that it possesses superior robustness of threshold compared to self-consistency. The findings are consistent in the zero-shot setting, with further details provided in Appendix D.

### 4.5.2 Dynamic Voting with Verifier

There are also multi-voting techniques to improve reasoning performance using additional verifiers, such as DIVERSE (Li et al., 2022). We conduct experiments with DIVERSE on the GSM8K dataset to assess the compatibility of Dynamic Voting with additional verifiers. As shown in Table 3, Dynamic Voting demonstrates similar efficiency to DIVERSE, that is, Dynamic Voting outperforms DIVERSE by a wide margin at low reasoning path usage and achieves comparable accuracies as DIVERSE with less than half the reasoning paths. This highlights the strong compatibility of Dynamic Voting when combined with other voting techniques.

### 4.5.3 Dynamic Voting on Other LLMs

Dynamic Voting is a technique of integration at the output side, with dynamic adjustment of the number of inferences, which can be used logically on all language models. We further experiment with

Openai's *code-davinci-002*[8] API, *GPT-4*[9] API, and LLaMA-7B (Touvron et al., 2023) on the GSM8K task in the few-shot setting. The performance of GPT-4 stands out with high accuracy and consistency, achieving an impressive accuracy rate of 95.2% when employing four votes in SC. Moreover, we observe no significant improvement in accuracy when increasing the number of votes from four to ten. Consequently, we consider four votes in the SC method as 100% path usage for our experiments with GPT-4, while 25% path usage indicates a single answer without voting. When conducting experiments with Code-davinci-002 and LLaMA-7B, we maintain the 100% path usage as forty votes. The results of the experiments are presented in Table 4. Dynamic Voting outperforms self-consistency significantly at low reasoning paths usage and achieves comparable accuracies to the best performance of self-consistency using fewer reasoning paths, which demonstrates the generalizability of Dynamic Voting across different models. It is noteworthy that achieving a target accuracy of 95% using GPT-4 incurs a cost exceeding $40 when employing SC, whereas the cost is less than $24 when utilizing Dynamic Voting.

### 4.5.4 Dynamic Voting Across Varied Prompts

Different prompts can lead to varying reasoning accuracies in large language models (Wang et al., 2022b,a; Zhang et al., 2022b). In this analysis, we aim to investigate the efficiency of Dynamic Voting when employed with different prompts. We compare the accuracies of Dynamic Voting and self-consistency on the GSM8K dataset at various paths usage levels, using another two distinct prompts, one generated by ChatGPT[10] (Prompt 2) and one written by the authors (Prompt 3). The whole prompts used can be found in Appendix E.

As shown in Table 5, Dynamic Voting con-

---

[8] https://platform.openai.com/docs/models/gpt-3-5.
[9] https://platform.openai.com/docs/models/gpt-4.
[10] https://chat.openai.com.

sistently achieves higher accuracies than self-consistency when the reasoning paths usage is below 50%. Moreover, Dynamic Voting always achieves accuracies comparable to self-consistency with only about 50% of the reasoning paths, regardless of the prompt used. These findings highlight the robustness of Dynamic Voting in maintaining high performance across diverse prompts.

# 5 Conclusion

To address the problem of computational resource wastage caused by multi-path voting using large language models, this paper proposes Dynamic Voting and designs Confidence-based and Percentage-based Dynamic Voting methods, which achieve comparable performance to self-consistency using significantly fewer reasoning paths. We demonstrate the effectiveness and efficiency of Dynamic Voting on Arithmetic Reasoning (GSM8K, SVAMP), Commensense Reasoning (CSQA, StaategyQA), and Symbolic Reasoning (LetterConcat) tasks using the GPT-3.5 model in few-shot and zero-shot settings. Dynamic Voting achieves comparable accuracies to self-consistency on all the five evaluated tasks with less than half of the reasoning paths. Moreover, Dynamic Voting performs more robustly for the selection of thresholds than self-consistency and demonstrates its generalizability combined with other voting techniques, different LLMs, and varied prompts. Our study provides valuable insights for developing efficient and effective reasoning methods with large language models. Future research can explore more sophisticated Dynamic Voting methods or explore the application of Dynamic Voting in other learning scenarios.

## Limitations

Dynamic Voting is an effective approach that can significantly reduce computational resource consumption when using large language models (LLMs) for multi-path voting to solve reasoning problems. Moreover, it has shown promising performance in both the few-shot and zero-shot settings. Nevertheless, we acknowledge five limitations of this approach.

Firstly, Dynamic Voting cannot calibrate the wrong answer to a problem that LLMs would not be able to solve. LLMs may stubbornly assume that certain incorrect facts are correct and repeatedly generate these in their reasoning. In such cases, Dynamic Voting can only end the path generation

process early without calibrating the answer to be correct.

Secondly, Confidence-based Dynamic Voting requires some prior knowledge of the likelihood of the LLMs solving the problem. An inappropriate threshold setting can still invalidate Dynamic Voting, for instance, if the model has a probability of generating a correct answer below 0.1 when faced with a difficult problem and we set the threshold to 0.9, this can degrade Dynamic Voting into self-consistency.

Thirdly, Percentage-based Dynamic Voting is not adept at handling individual questions, which presents a challenge for deploying applications that handle single instance requests in real time.

Fourthly, due to the inaccessibility of other large language models, Dynamic Voting has been tested only on the GPT series models and LLaMA-7B. Yet, this limitation is currently widespread in works that explore LLMs.

Lastly, Dynamic Voting has only been tested on English reasoning tasks with specific answers and is not suited to natural language generation tasks such as summarization and translation. However, it is essential to note that this limitation is not unique to Dynamic Voting but is inherent in all current multi-path voting methods.

In summary, while Dynamic Voting is a valuable approach for reducing computational resource consumption in multi-path voting, it is crucial to consider these limitations carefully to optimize its performance and applicability in real-world scenarios. Subsequent efforts will center on ameliorating these limitations and extending the scope of Dynamic Voting scenarios.

## Ethics Statement

We hereby affirm that all co-authors of this paper are cognizant of and uphold the ACL Code of Ethics. In this study, we present an effective approach that significantly curtails resource consumption while using large models for multi-path voting to resolve reasoning problems. Our contribution concentrates on the methodology rather than the development of data and language models, which does not raise any ethical issues. Nonetheless, the publicly accessible datasets and pre-trained models employed in this research may harbor negative effects, such as gender and religious bias. As a result, we urge other researchers to exercise caution when employing our methods and these data.

## Acknowledgments

This work is supported by the Fundamental Research Funds for the Central Universities under Grant 1082204112364 and the Key Program of the National Science Foundation of China under Grant 61836006.

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

## A  Resource Wastage in Multi-path Voting

Multi-path voting methods like self-consistency emerge as potent techniques for enhancing the reasoning accuracy of large language models. However, the drawback of these methods lies in the inefficient utilization of computational resources due to the generation of an excessive number of reasoning paths for each problem.

We extensively explore the computational resource wastage in multi-path voting, using self-consistency on GPT-3.5 across five distinct datasets as an example. When voting with 3 paths, we find that there are 39.4%, 57.5%, 47.2%, 33.8%, and 59.0% of the questions on the GSM8K, SVAMP, CSQA, StrategyQA, and LetterConcat datasets obtain unanimous votes, respectively. Then we increase the number of paths for voting on the questions receiving unanimous votes, and the results are presented in Table 6. It becomes evident that augmenting the number of paths does not yield a significant enhancement in accuracy for these questions. In particular, a degradation in accuracy can be observed on StrategyQA. This observation implies that allocating substantial inference computational resources to nearly half of the problems in the datasets is redundant, as an investment of over ten times the computational resources results in less than 1% improvement in accuracy. To mitigate the issue, Dynamic Voting applies early exiting for problems that demonstrate high consistency in voting, thereby curbing unnecessary computations.

| Paths | GSM8K | SVAMP | CSQA | StrategyQA | LetterConcat |
|-------|-------|-------|------|------------|--------------|
| 3 | 97.1 | 95.7 | 89.6 | 87.2 | 94.9 |
| 5 | 97.1 | 95.7 | 89.6 | 87.2 | 94.9 |
| 10 | 97.3 | 95.8 | 89.6 | 87.2 | 95.1 |
| 20 | 97.3 | 95.7 | 89.9 | 87.2 | 95.8 |
| 40 | 97.3 | 96.5 | 89.9 | 86.9 | 95.8 |

Table 6: Self-consistency accuracies on questions that get unanimous votes with 3 paths.

## B  Searched Thresholds

The searched thresholds are shown in Table 7.

## C  Results under Different Path Usages

The experimental results under different path usages are shown in Figure 5. It is clear that Dynamic Voting consistently achieves higher accuracies than self-consistency on all the five evaluated datasets when paths are limited to under 50%. Except for

| Task | Method | fewshot | | | | zeroshot | | | |
|---|---|---|---|---|---|---|---|---|---|
| | | $\leq 25\%$ | $\leq 50\%$ | $\leq 75\%$ | $\leq 100\%$ | $\leq 25\%$ | $\leq 50\%$ | $\leq 75\%$ | $\leq 100\%$ |
| GSM8K | SC | 10 | 20 | 30 | 40 | 10 | 20 | 30 | 40 |
| | CDV | 0.40 | 0.70 | 0.95 | 0.95 | 0.35 | 0.55 | 0.70 | 0.95 |
| | PDV | 17 | 7 | 3 | 1 | 16 | 7 | 4 | 1 |
| SVAMP | SC | 10 | 20 | 30 | 40 | 10 | 20 | 30 | 40 |
| | CDV | 0.55 | 0.70 | 0.95 | 0.95 | 0.45 | 0.65 | 0.95 | 0.95 |
| | PDV | 16 | 6 | 4 | 1 | 17 | 7 | 3 | 1 |
| CSQA | SC | 10 | 20 | 30 | 40 | 10 | 20 | 30 | 40 |
| | CDV | 0.55 | 0.80 | 0.95 | 0.95 | 0.10 | 0.30 | 0.45 | 0.95 |
| | PDV | 21 | 6 | 3 | 1 | 20 | 7 | 3 | 1 |
| StrategyQA | SC | 10 | 20 | 30 | 40 | 10 | 20 | 30 | 40 |
| | CDV | 0.40 | 0.70 | 0.90 | 0.95 | 0.40 | 0.55 | 0.80 | 0.95 |
| | PDV | 20 | 9 | 3 | 1 | 20 | 6 | 4 | 1 |
| LetterConcat | SC | 10 | 20 | 30 | 40 | 10 | 20 | 30 | 40 |
| | CDV | 0.70 | 0.95 | 0.95 | 0.95 | 0.20 | 0.30 | 0.40 | 0.95 |
| | PDV | 17 | 7 | 3 | 1 | 17 | 6 | 3 | 1 |

Table 7: The thresholds used under different resource consumption. The thresholds in SC means the number of used votes. The thresholds in CSV and PDV means consistency thresholds $t_c$ and percentage thresholds $t_p$, respectively. These thresholds are searched via a validation set containing 100 randomly chosen samples from training sets. The threshold value indicates that when the threshold is applied by the method, less than the corresponding percentage of paths are used on the validation set. For instance, 0.40 in the second row (GSM8K, CDV, fewshot, $\leq 25\%$) means that when the threshold is set to 0.40, CDV in the few-shot setting on GSM8K uses less than 25% of the paths compared with SC which using 40 votes per question.

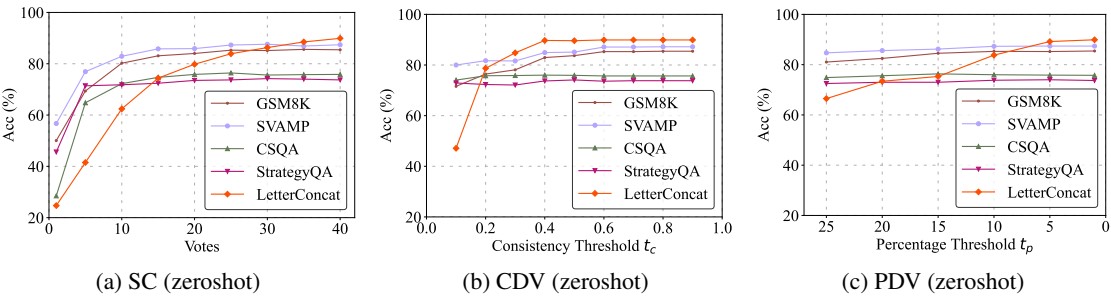

| (a) SC (zeroshot) | (b) CDV (zeroshot) | (c) PDV (zeroshot) |

Figure 4: The zeroshot results of Self-consistency and Dynamic Voting under different thresholds.

the few-shot setting on StrategyQA and LetterConcat, Dynamic Voting outperforms self-consistency using fewer paths.

## D   Zero-shot Results with Different Thresholds

The experimental results with different thresholds in the zero-shot setting are shown in Figure 4. The results in the zero-shot setting lead to similar conclusions as those from the results in the few-shot setting in Figure 3. That is, Dynamic Voting exhibits greater robustness of thresholds compared to self-consistency.

## E   Prompts and Instructions

The prompts and instructions used in this paper are shown from Table 8 through Table 13. All of the prompts 1 are rewritten from the prompts in Wang et al. (2022b) into dialogue form. On GSM8K, prompt 2 is generated by ChatGPT[11], and prompt 3 is written by the authors of this paper. All experiments in this paper are done based on prompts 1 except for the analysis on prompts in Section 4.5.4.

---

[11] https://chat.openai.com.

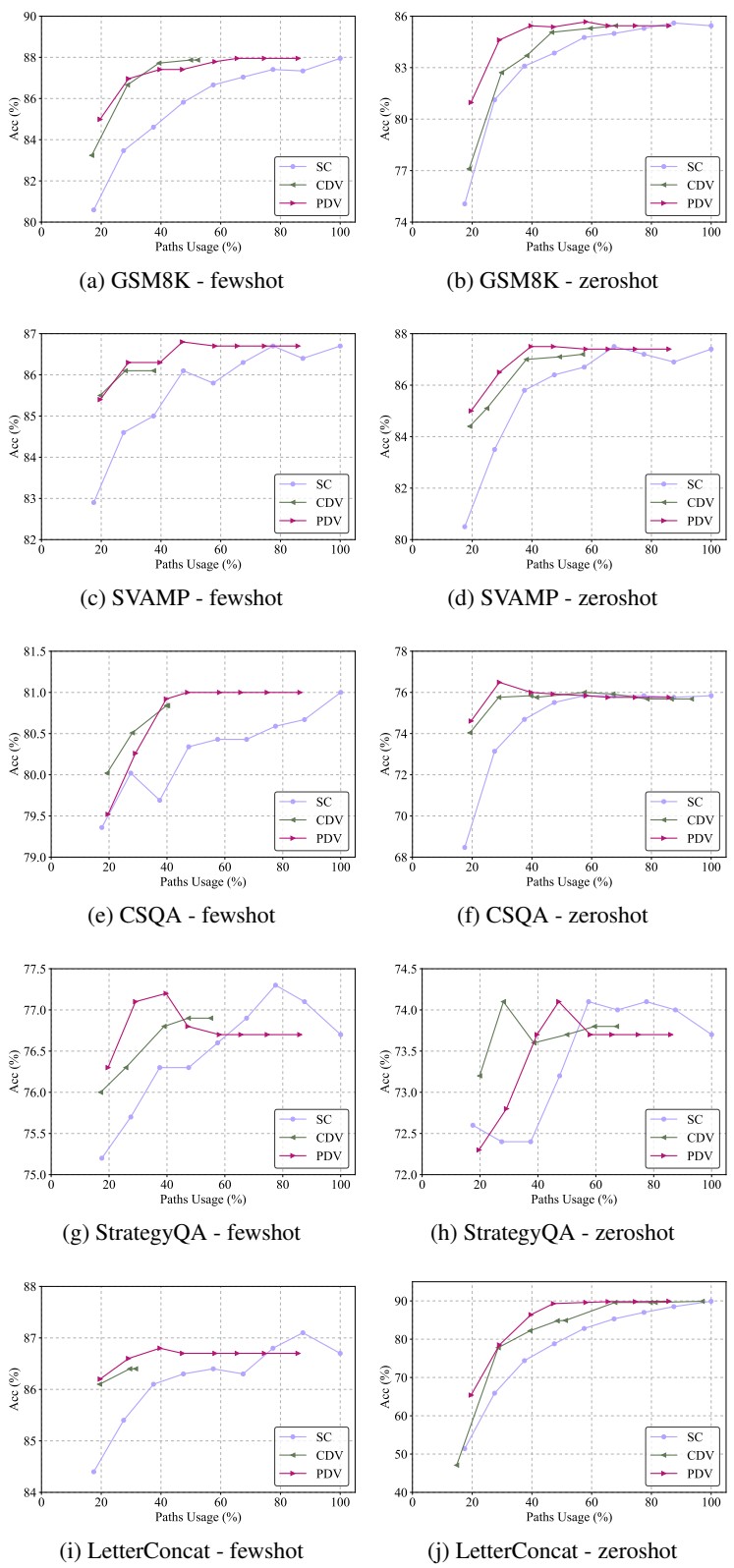

Figure 5: The results of Self-consistency and Dynamic Voting under different reasoning paths usages.

**assistant:** We start with 15 trees. Later we have 21 trees. The difference must be the number of trees they planted. So, they must have planted 21 - 15 = 6 trees. The answer is 6.

**assistant:** There are 3 cars in the parking lot already. 2 more arrive. Now there are 3 + 2 = 5 cars. The answer is 5.

**assistant:** Leah had 32 chocolates and Leah's sister had 42. That means there were originally 32 + 42 = 74 chocolates. 35 have been eaten. So in total they still have 74 - 35 = 39 chocolates. The answer is 39.

**assistant:** Jason had 20 lollipops. Since he only has 12 now, he must have given the rest to Denny. The number of lollipops he has given to Denny must have been 20 - 12 = 8 lollipops. The answer is 8.

**assistant:** He has 5 toys. He got 2 from mom, so after that he has 5 + 2 = 7 toys. Then he got 2 more from dad, so in total he has 7 + 2 = 9 toys. The answer is 9.

**assistant:** Michael initially had 58 balls. He lost 23 on Tuesday, so after that he has 58 - 23 = 35 balls. On Wednesday he lost 2 more so now he has 35 - 2 = 33 balls. The answer is 33.

**assistant:** She bought 5 bagels for $3 each. This means she spent 5 * $3 = $15 on the bagels. She had $23 in beginning, so now she has $23 - $15 = $8. The answer is 8.

Table 8: The prompt and instruction used on GSM8K and SVAMP. The notation [question] refers to the question that is being evaluated.

*GSM8K / SVAMP Prompt 2:*

**system:** You are a helpful assistant good at mathematical reasoning.

**user:** There are 15 trees in the grove. Grove workers will plant trees in the grove today. After they are done, there will be 21 trees. How many trees did the grove workers plant today?

**assistant:** To find out how many trees the grove workers planted, we need to subtract the initial number of trees from the final number of trees: Number of trees planted = final number of trees - initial number of trees = 21 - 15 = 6. Therefore, the grove workers planted 6 trees today. The answer is 6.

**user:** If there are 3 cars in the parking lot and 2 more cars arrive, how many cars are in the parking lot?

**assistant:** If there are 3 cars in the parking lot and 2 more cars arrive, the total number of cars in the parking lot is: 3 + 2 = 5. Therefore, there are 5 cars in the parking lot. The answer is 5.

**user:** Leah had 32 chocolates and her sister had 42. If they ate 35, how many pieces do they have left in total?

**assistant:** Leah had 32 chocolates and her sister had 42, so they had a total of: 32 + 42 = 74 chocolates. If they ate 35 chocolates, the number of chocolates they have left is: 74 - 35 = 39 chocolates. Therefore, they have 39 chocolates left in total. The answer is 39.

**user:** Jason had 20 lollipops. He gave Denny some lollipops. Now Jason has 12 lollipops. How many lollipops did Jason give to Denny?

**assistant:** To find out how many lollipops Jason gave to Denny, we need to subtract the final number of lollipops Jason has from the initial number of lollipops he had: Number of lollipops given to Denny = initial number of lollipops - final number of lollipops = 20 - 12 = 8. Therefore, Jason gave 8 lollipops to Denny. The answer is 8.

**user:** Shawn has five toys. For Christmas, he got two toys each from his mom and dad. How many toys does he have now?

**assistant:** Shawn has five toys initially. For Christmas, he got two toys each from his mom and dad, so he got a total of: 2 (toys from mom) + 2 (toys from dad) = 4 toys. Adding the four toys from Christmas to his initial five toys, we get: 5 + 4 = 9. Therefore, Shawn has 9 toys now. The answer is 9.

**user:** There were nine computers in the server room. Five more computers were installed each day, from monday to thursday. How many computers are now in the server room?

**assistant:** If there were 9 computers to start with and 5 more were added each day from Monday to Thursday, that means 4 days have passed since Monday to Thursday. So, the number of additional computers added would be: 5 + 5 + 5 + 5 = 20. Therefore, the total number of computers in the server room now would be: 9 + 20 = 29. Hence, there are 29 computers in the server room now. The answer is 29.

**user:** Michael had 58 golf balls. On tuesday, he lost 23 golf balls. On wednesday, he lost 2 more. How many golf balls did he have at the end of wednesday?

**assistant:** Michael had 58 golf balls. On Tuesday, he lost 23 golf balls, which leaves him with: 58 - 23 = 35 golf balls. On Wednesday, he lost 2 more golf balls, which leaves him with: 35 - 2 = 33 golf balls. Therefore, Michael had 33 golf balls at the end of Wednesday. The answer is 33.

**user:** Olivia has $23. She bought five bagels for $3 each. How much money does she have left?

**assistant:** Olivia had $23 to start with. She bought 5 bagels at $3 each, which is a total cost of: 5 x $3 = $15. Therefore, after buying the bagels, Olivia has: $23 - $15 = $8 left. So, Olivia has $8 left after buying bagels. The answer is 8.

**user:** [question]

Table 9: The prompt 2 used on GSM8K and SVAMP.

*GSM8K / SVAMP Prompt 3:*

**system:** You are a helpful assistant good at mathematical reasoning.

**user:** There are 15 trees in the grove. Grove workers will plant trees in the grove today. After they are done, there will be 21 trees. How many trees did the grove workers plant today?

**assistant:** Step 1: We start with 15 trees. Later we have 21 trees. The difference must be the number of trees they planted. So, they must have planted 21 - 15 = 6 trees. Step 2: The answer is 6.

**user:** If there are 3 cars in the parking lot and 2 more cars arrive, how many cars are in the parking lot?

**assistant:** Step 1: There are 3 cars in the parking lot already. 2 more arrive. Now there are 3 + 2 = 5 cars. Step 2: The answer is 5.

**user:** Leah had 32 chocolates and her sister had 42. If they ate 35, how many pieces do they have left in total?

**assistant:** Step 1: Leah had 32 chocolates and Leah's sister had 42. That means there were originally 32 + 42 = 74 chocolates. Step 2: 35 have been eaten. So in total they still have 74 - 35 = 39 chocolates. Step 3: The answer is 39.

**user:** Jason had 20 lollipops. He gave Denny some lollipops. Now Jason has 12 lollipops. How many lollipops did Jason give to Denny?

**assistant:** Step 1: Jason had 20 lollipops. Since he only has 12 now, he must have given the rest to Denny. The number of lollipops he has given to Denny must have been 20 - 12 = 8 lollipops. Step 2: The answer is 8.

**user:** Shawn has five toys. For Christmas, he got two toys each from his mom and dad. How many toys does he have now?

**assistant:** Step 1: He has 5 toys. He got 2 from mom, so after that he has 5 + 2 = 7 toys. Step 2: Then he got 2 more from dad, so in total he has 7 + 2 = 9 toys. Step 3: The answer is 9.

**user:** There were nine computers in the server room. Five more computers were installed each day, from monday to thursday. How many computers are now in the server room?

**assistant:** Step 1: There are 4 days from monday to thursday. 5 computers were added each day. That means in total 4 * 5 = 20 computers were added. Step 2: There were 9 computers in the beginning, so now there are 9 + 20 = 29 computers. Step 3: The answer is 29.

**user:** Michael had 58 golf balls. On tuesday, he lost 23 golf balls. On wednesday, he lost 2 more. How many golf balls did he have at the end of wednesday?

**assistant:** Step 1: Michael initially had 58 balls. He lost 23 on Tuesday, so after that he has 58 - 23 = 35 balls. Step 2: On Wednesday he lost 2 more so now he has 35 - 2 = 33 balls. Step 3: The answer is 33.

**user:** Olivia has $23. She bought five bagels for $3 each. How much money does she have left?

**assistant:** Step 1: She bought 5 bagels for $3 each. This means she spent 5 * $3 = $15 on the bagels. Step 2: She had $23 in beginning, so now she has $23 - $15 = $8. Step 3: The answer is 8.

**user:** [question]

Table 10: The prompt 3 used on GSM8K and SVAMP.

## CSQA Prompt 1 - Fewshot Setting:

**system:** You are a helpful assistant good at commonsense reasoning.

**user:** What do people use to absorb extra ink from a fountain pen?

Answer Choices: (a) shirt pocket (b) calligrapher's hand (c) inkwell (d) desk drawer (e) blotter

**assistant:** The answer must be an item that can absorb ink. Of the above choices, only blotters are used to absorb ink. So the answer is (e).

**user:** What home entertainment equipment requires cable?

Answer Choices: (a) radio shack (b) substation (c) television (d) cabinet

**assistant:** The answer must require cable. Of the above choices, only television requires cable. So the answer is (c).

**user:** The fox walked from the city into the forest, what was it looking for?

Answer Choices: (a) pretty flowers (b) hen house (c) natural habitat (d) storybook

**assistant:** The answer must be something in the forest. Of the above choices, only natural habitat is in the forest. So the answer is (c).

**user:** Sammy wanted to go to where the people were. Where might he go?

Answer Choices: (a) populated areas (b) race track (c) desert (d) apartment (e) roadblock

**assistant:** The answer must be a place with a lot of people. Of the above choices, only populated areas have a lot of people. So the answer is (a).

**user:** Where do you put your grapes just before checking out?

Answer Choices: (a) mouth (b) grocery cart (c) super market (d) fruit basket (e) fruit market

**assistant:** The answer should be the place where grocery items are placed before checking out. Of the above choices, grocery cart makes the most sense for holding grocery items. So the answer is (b).

**user:** Google Maps and other highway and street GPS services have replaced what?

Answer Choices: (a) united states (b) mexico (c) countryside (d) atlas

**assistant:** The answer must be something that used to do what Google Maps and GPS services do, which is to give directions. Of the above choices, only atlases are used to give directions. So the answer is (d).

**user:** Before getting a divorce, what did the wife feel who was doing all the work?

Answer Choices: (a) harder (b) anguish (c) bitterness (d) tears (e) sadness

**assistant:** The answer should be the feeling of someone getting divorced who was doing all the work. Of the above choices, the closest feeling is bitterness. So the answer is (c).

**user:** [question]

## CSQA Instruction - Zeroshot Setting:

**system:** You are a helpful assistant good at commonsense reasoning.

**user:** [question] Solve this problem in 80 words and end with 'the answer is x.'

Table 11: The prompt and instruction used on CSQA.

*StrategyQA Prompt 1 - Fewshot Setting:*

**system:** You are a helpful assistant good at commonsense reasoning.

**user:** Yes or no: Do hamsters provide food for any animals?

**assistant:** Hamsters are prey animals. Prey are food for predators. Thus, hamsters provide food for some animals. So the answer is yes.

**user:** Yes or no: Could Brooke Shields succeed at University of Pennsylvania?

**assistant:** Brooke Shields went to Princeton University. Princeton University is about as academically rigorous as the University of Pennsylvania. Thus, Brooke Shields could also succeed at the University of Pennsylvania. So the answer is yes.

**user:** Yes or no: Hydrogen's atomic number squared exceeds number of Spice Girls?

**assistant:** Hydrogen has an atomic number of 1. 1 squared is 1. There are 5 Spice Girls. Thus, Hydrogen's atomic number squared is less than 5. So the answer is no.

**user:** Yes or no: Is it common to see frost during some college commencements?

**assistant:** College commencement ceremonies can happen in December, May, and June. December is in the winter, so there can be frost. Thus, there could be frost at some commencements. So the answer is yes.

**user:** Yes or no: Could a llama birth twice during War in Vietnam (1945-46)?

**assistant:** The War in Vietnam was 6 months. The gestation period for a llama is 11 months, which is more than 6 months. Thus, a llama could not give birth twice during the War in Vietnam. So the answer is no.

**user:** Yes or no: Would a pear sink in water?

**assistant:** The density of a pear is about 0.6g/cm^3, which is less than water. Objects less dense than water float. Thus, a pear would float. So the answer is no.

**user:** [question]

*StrategyQA Instruction - Zeroshot Setting:*

**system:** You are a helpful assistant good at commonsense reasoning.

**user:** [question] Solve this problem in 80 words and end with 'the answer is x.'

Table 12: The prompt and instruction used on StrategyQA.

---

*LetterConcat Prompt 1 - Fewshot Setting:*

**system:** You are a helpful assistant good at symbolic reasoning.

**user:** Take the last letters of the words in "Elon Musk" and concatenate them.

**assistant:** The last letter of "Elon" is "n". The last letter of "Musk" is "k". Concatenating them is "nk". The answer is nk.

**user:** Take the last letters of the words in "Larry Page" and concatenate them.

**assistant:** The last letter of "Larry" is "y". The last letter of "Page" is "e". Concatenating them is "ye". The answer is ye.

**user:** Take the last letters of the words in "Sergey Brin" and concatenate them.

**assistant:** The last letter of "Sergey" is "y". The last letter of "Brin" is "n". Concatenating them is "yn". The answer is yn.

**user:** Take the last letters of the words in "Bill Gates" and concatenate them.

**assistant:** The last letter of "Bill" is "l". The last letter of "Gates" is "s". Concatenating them is "ls". The answer is ls.

**user:** [question]

---

*LetterConcat Instruction - Zeroshot Setting:*

**system:** You are a helpful assistant good at symbolic reasoning.

**user:** [question] Solve this problem in 80 words and end with 'the answer is x.'

---

Table 13: The prompt and instruction used on LetterConcat.