# OpenReview forum: "Dynamic Voting for Efficient Reasoning in Large Language Models"
_EMNLP/2023/Conference — EMNLP 2023 Findings_

### Official Review · Reviewer_nuFX · 2023-08-02

**Soundness:** 4

**Excitement:**

3: Ambivalent: It has merits (e.g., it reports state-of-the-art results, the idea is nice), but there are key weaknesses (e.g., it describes incremental work), and it can significantly benefit from another round of revision. However, I won't object to accepting it if my co-reviewers champion it.

**Paper Topic And Main Contributions:**

Self-consistency approach to guide LLMs decoding showed significant performance improvements on various language tasks. Authors propose to improve the efficiency of this method by introducing early stopping. In particular, authors propose to use dynamic voting: instead of sampling large fixed number (usually ~40) of candidates, authors generate a subsamples and stop generations once desired level of consistency is achieved. Authors compare confidence-based and percentage-based stopping criteria, and show that the proposed approach achieves compatible performance with self-consistency, while improving efficiency in terms of number of generations.

**Questions For The Authors:**

Suggestions:
- It would be interesting to see other stopping criteria explored, beyond simple consistency-based voting.
- To improve reproducibility, please consider using opensourced LLMs

**Reasons To Accept:**

Authors propose an interesting approach to improve the efficiency of the existing method. The paper is well-written, with detailed explanations of the approach. Authors perform experiments on a wide variety of tasks and show consistent results.

**Reasons To Reject:**

The paper presents a minor improvement over the existing approach. While it is important to improve decoding efficiency, it is not clear how much gains it will give in terms of improving time or memory consumption. For large enough batch sizes, reducing number of candidates to 75% (to get same performance, I'm looking at Table 1 for the reference), or from 40 to 30 candidates, could bring limited improvements.

**Reproducibility:**

4: Could mostly reproduce the results, but there may be some variation because of sample variance or minor variations in their interpretation of the protocol or method.

**Reviewer Confidence:**

5: Positive that my evaluation is correct. I read the paper very carefully and I am very familiar with related work.

---

> ### Author Rebuttal · Authors · 2023-08-25
>
> We are glad you find our approach interesting and appreciate the insightful suggestions.
>
> We agree that reducing the batch size from 40 to 30 does not seem to reduce the memory and time overhead very significantly. But our approach shows the potential to go beyond that. As shown in Table 1, our method Confidence-based Dynamic Voting in the few-shot setting achieves an average accuracy of 83.8 using 37.6% of the paths as SC hits 83.9 using 100% of paths, which means that it is possible to process double or even triple the number of samples in one batch with little to no performance degradation when using Dynamic Voting. This can result in substantial cost savings, particularly for organizations with limited resources. In addition, the most powerful GPT4 API currently charges by the number of tokens. As an example, Percentage-based Dynamic Voting outperforms SC on the LetterConcat task with 24.8% path usage. Completing the LetterConcat test set using GPT4 utilizing SC would cost roughly \\$307 (1000 samples * 40 paths/sample * 128 tokens/path * \\$0.06/1000 tokens = \\$307.2) while 24.8% means \\$76. We believe our approach offers an attractive solution for researchers and organizations seeking to optimize their resource utilization.
>
>  Regarding the suggestions:
>
> 1. We have evaluated the effectiveness of our method in combination with Verifier, a model that assigns scores to each path and halts inference when multiple high-scoring paths converge on the same reasoning result (Section 4.5.2). The results presented in Table 3 clearly demonstrate the successful integration of Dynamic Voting and Verifier, yielding a high level of accuracy with significantly fewer paths. We acknowledge the potential benefits of incorporating better stopping criteria in our research and will make this a key focus in our future efforts.
> 2. We have evaluated our method and demonstrated its effectiveness on an open-source model, namely LLaMA (Touvron et al., 2023), in Section 4.5.3. To maintain consistency with prior studies (Wang et al., 2022b; Li et al., 2022; Wang et al., 2022a), we opt to employ the GPT API in our primary experiments. However, we acknowledge the significance of open-source LLMs and will further explore their potential in future investigations.

---

### Official Review · Reviewer_NY3h · 2023-08-05

**Soundness:** 2

**Excitement:**

2: Mediocre: This paper makes marginal contributions (vs non-contemporaneous work), so I would rather not see it in the conference.

**Paper Topic And Main Contributions:**

Dynamic Voting for Efficient Reasoning in LLMs

In this paper the authors describe two simple checks to reduce the number of alternative generative paths in LLMs while still enabling accirate responses. One check looks at the consistency between different paths and the other returns responses to queries in a top range of consistency. They show using 6 tasks against one baseline that their approach outperforms the baseline.

**Reasons To Accept:**

+ very simple approaches to reducing computational cost while maintaining accuracy in LLM reasoning
+ mostly well written paper
+ many detailed experiments to illustrate different aspects of the approach

**Reasons To Reject:**

- the methods are rather weak. Significance/Impact of the work is likely limited.
- some important details are unclear, such as how exactly the different paths are generated (the two lines in 4.3 are not enough); how is it ensured the paths are sufficiently different; what is done if they are simple variations of each other. Voting generally will not work well if the elements being voted are correlated significantly.
- lacks comparison to alternative strategies to reduce computation/reasoning.

**Reproducibility:**

3: Could reproduce the results with some difficulty. The settings of parameters are underspecified or subjectively determined; the training/evaluation data are not widely available.

**Reviewer Confidence:**

3: Pretty sure, but there's a chance I missed something. Although I have a good feel for this area in general, I did not carefully check the paper's details, e.g., the math, experimental design, or novelty.

---

> ### Author Rebuttal · Authors · 2023-08-28
>
> Thank you for taking the time to review our manuscript. We hope that our response may alleviate your doubt.
>
> $\text{\color{blue}Reason 1: The methods are rather weak. Significance/Impact of the work is likely limited.}$
>
> Answer 1: Existing LLMs all suffer from hallucination and factual errors, despite the powerful GPT-4. Multipath voting is an effective way to improve the accuracy of reasoning, which has been well demonstrated in previous works (Wang et al., 2022b; Li et al., 2022; Wang et al., 2022a). However, multipath voting brings multiplicative resource consumption as LLMs generate multiple reasoning paths per question. We apply the idea of early exiting to multipath voting and propose Dynamic Voting to achieve comparable performance with significantly fewer paths.
>
> In particular, Percentage-based Dynamic Voting outperforms SC (vanilla multipath voting) on the LetterConcat task with 24.8% path usage. As an example, completing the LetterConcat test set using GPT4 utilizing SC would cost roughly \\$307 (1000 samples * 40 paths/sample * 128 tokens/path * \\$0.06/1000 tokens = \\$307.2) while 24.8% means \\$76. Although our approach and intuition may appear relatively simple, we firmly believe that Dynamic Voting represents a highly effective and meaningful technique in saving resources and money.
>
> $\text{\color{blue}Reason 2: How exactly the different paths are generated is not detailed.}$
>
> Answer 2: We state that we use temperature sampling in decoding to obtain diverse reasoning paths and give the used temperature and top-p in lines 362-366. They are the only two parameters to obtain diverse generations in OpenAI GPT API. Previous works (Wang et al., 2022b; Li et al., 2022; Wang et al., 2022a) have demonstrated the effectiveness of temperature sampling in multipath voting.
>
> $\text{\color{blue}Reason 3: How is it ensured the paths are sufficiently different; What is done if they are simple variations of each other.}$
>
> Answer 3: Previous works (Wang et al., 2022b; Li et al., 2022; Wang et al., 2022a) have addressed these concerns and they are not the focus nor the contribution of our work.
>
> $\text{\color{blue}Reason 4: Lacks comparison to alternative strategies to reduce computation/reasoning.}$
>
> Answer 4: As of the submission deadline, this work is the first and only work that studies resource conservation in multipath reasoning with LLMs. Other than the baseline we used (Wang et al., 2022b), we do not realize any other suitable baseline for comparison.

---

### Official Review · Reviewer_TXTM · 2023-08-06

**Soundness:** 3

**Excitement:**

3: Ambivalent: It has merits (e.g., it reports state-of-the-art results, the idea is nice), but there are key weaknesses (e.g., it describes incremental work), and it can significantly benefit from another round of revision. However, I won't object to accepting it if my co-reviewers champion it.

**Paper Topic And Main Contributions:**

This paper proposes a novel multi-path voting technique called Dynamic Voting, which effectively reduces the number of reasoning paths during multi-path voting while preserving accuracies by applying early exit. The authors introduce two Dynamic Voting strategies, Confidence-based Dynamic Voting (CDV) and Percentage-based Dynamic Voting (PDV), and evaluate their performance in arithmetic, commonsense, and symbolic reasoning tasks under few-shot and zero-shot settings and perform additional analysis to better position their techniques in the literature.

**Questions For The Authors:**

1. In Sec 4.2, can the authors state exactly which gpt3.5 version is used for reproducibility purposes?
2. gpt3.5 is much weaker than gpt4, and i would imagine gpt4 can use much fewer calls to solve the same problems. It would be interesting to show, given the same money budget, can gpt-4 perform better or gpt3.5 perform better?  If using gpt3.5 with dynamic voting ending up using more money, it's hard for me to imagine practitioners actually use this technique.  Alternatively, the authors could show gpt4 can also benefit from this technique?
3. The main experiment table is confusing to read.  First, I don't understand why self-consistency also has the budget constraint?  Second, I don't understand when numbers are bolded / underlined.  Can the authors just be explicit about which specific baseline number they are comparing to?

**Reasons To Accept:**

1. The early exiting idea + voting idea is simple and can save computation
2. The evaluation is pretty comprehensive w.r.t. datasets and settings
3. The analysis is interesting and comprehensive
4. The writing is pretty clear

**Reasons To Reject:**

1. The main experiment table is confusing, making people hard to understand the actual empirical performance of the proposed approach.

**Reproducibility:**

3: Could reproduce the results with some difficulty. The settings of parameters are underspecified or subjectively determined; the training/evaluation data are not widely available.

**Reviewer Confidence:**

4: Quite sure. I tried to check the important points carefully. It's unlikely, though conceivable, that I missed something that should affect my ratings.

---

> ### Author Rebuttal · Authors · 2023-08-25
>
> Thank you for your valuable comment regarding the clarity of the main experiment table. The design of Table 1 is inspired by Table 3 in Kaya et al. (2019). We opt for this as it also focuses on early exiting to conserve resources. In addition to the table, we have included Figure 5, a line graph that visually represents the accuracies of Dynamic Voting and SC under different path usages.
>
> To address your concerns, we would like to provide further clarification on the content and structure of the table.
>  - It is worth noting that SC utilizes 40 paths per question for voting, but we acknowledge that this value is not an infallible hyperparameter. To ensure a fair comparison of accuracies under different path usage conditions, we introduce usage constraints to SC. For instance, a usage constraint of 25% signifies that 10 paths are used per question, while a constraint of 50% corresponds to 20 paths, and so forth.
>  - The percentages presented at the top of the table indicate the path usage constraints corresponding to the accuracies displayed in each column.
>  - The thresholds for Dynamic Voting are determined based on the holdout dev set, and we indicate the actual path usages on the test set in brackets.
>  - To highlight the superior accuracies of Dynamic Voting compared to SC under the same resource constraints, we denote the former in bold when they exceed the latter. Moreover, if the accuracies of Dynamic Voting match those of SC(40 votes), we underline them.
> We will try to optimize the presentation and caption of the table to enhance the clarity and comprehensibility of our work.
>
> Regarding the Questions:
>
>  - To Q1: As mentioned in lines 122-124 and the accompanying footnote, we utilize GPT-3.5-turbo-0301 for our experiments. To enhance clarity, we will explicitly restate this information in section 4.2 to ensure reproducibility.
>  - To Q2: We conduct an experiment using GPT-4 on GSM8K and find that Dynamic Voting works well with GPT-4. Our results demonstrate that GPT-4 achieves high accuracy and consistency, with an accuracy rate of 95.2% when using four votes in SC. Additionally, we observe no further accuracy improvements from four to ten votes. Therefore we use 4 votes on SC as 100% path usage. Confidence-based Dynamic Voting achieves a 95.2% accuracy rate with only 53.9% of the path usage. Percentage-based Dynamic Voting achieves a 95.4% accuracy rate with 59.7% of the path usage. For the API calls on this one task alone, SC costs us about \\$40, while CDV and PDV cost \\$22 and \\$24, respectively. We will include these findings in the next version of our manuscript.
>  - To Q3: Please refer to the response above.

---

### Meta-Review · Area_Chair_LMSA · 2023-09-17

**Recommendation:** 3

**Metareview:**

The paper introduces a multi-path voting technique called Dynamic Voting, specifically Confidence-based Dynamic Voting (CDV) and Percentage-based Dynamic Voting (PDV). The technique is designed to reduce the number of reasoning paths during multi-path voting while maintaining accuracy through early exit strategies. The authors evaluate CDV and PDV on various reasoning tasks under few-shot and zero-shot settings and provide comprehensive analyses.

Reviewers appreciate the innovative approach of using Dynamic Voting with early exit strategies to reduce computation while maintaining accuracy. This technique is seen as a valuable contribution to the field. The paper is commended for its comprehensive evaluation, including a wide range of datasets and settings. This thoroughness adds to the credibility of the proposed technique. The clarity of writing is acknowledged as a positive aspect of the paper, making it easy for readers to understand the proposed approach and its results. Some reviewers express concerns about the limited improvement achieved by the proposed method over existing approaches. In particular, the concerns centered around a minor improvement in terms of efficiency, and it is noted that the gains in terms of time or memory consumption may be limited, especially for larger batch sizes.

The paper presents an innovative approach to improving efficiency in LM decoding through Dynamic Voting with early exit strategies. The approach is novel and well-explained, however further clarification regarding the practical benefits, especially in terms of resource consumption, could strengthen the paper's contribution.

---

### Decision · Program_Chairs · 2023-10-07

**Decision:**

Accept-Findings

**Comment:**

The paper introduces a multi-path voting technique called Dynamic Voting, specifically Confidence-based Dynamic Voting (CDV) and Percentage-based Dynamic Voting (PDV). The technique is designed to reduce the number of reasoning paths during multi-path voting while maintaining accuracy through early exit strategies. The authors evaluate CDV and PDV on various reasoning tasks under few-shot and zero-shot settings and provide comprehensive analyses.

Reviewers appreciate the innovative approach of using Dynamic Voting with early exit strategies to reduce computation while maintaining accuracy. This technique is seen as a valuable contribution to the field. The paper is commended for its comprehensive evaluation, including a wide range of datasets and settings. This thoroughness adds to the credibility of the proposed technique. The clarity of writing is acknowledged as a positive aspect of the paper, making it easy for readers to understand the proposed approach and its results. Some reviewers express concerns about the limited improvement achieved by the proposed method over existing approaches. In particular, the concerns centered around a minor improvement in terms of efficiency, and it is noted that the gains in terms of time or memory consumption may be limited, especially for larger batch sizes.

The paper presents an innovative approach to improving efficiency in LM decoding through Dynamic Voting with early exit strategies. The approach is novel and well-explained, however further clarification regarding the practical benefits, especially in terms of resource consumption, could strengthen the paper's contribution.